# Western Diet and Cognitive Decline: A Hungarian Perspective—Implications for the Design of the Semmelweis Study

**DOI:** 10.3390/nu17152446

**Published:** 2025-07-27

**Authors:** Andrea Lehoczki, Tamás Csípő, Ágnes Lipécz, Dávid Major, Vince Fazekas-Pongor, Boglárka Csík, Noémi Mózes, Ágnes Fehér, Norbert Dósa, Dorottya Árva, Kata Pártos, Csilla Kaposvári, Krisztián Horváth, Péter Varga, Mónika Fekete

**Affiliations:** 1Institute of Preventive Medicine and Public Health, Semmelweis University, 1085 Budapest, Hungary; ceglediandi@freemail.hu (A.L.); csipo.tamas@semmelweis.hu (T.C.); lipecz.agnes@semmelweis.hu (Á.L.); major.david@semmelweis.hu (D.M.); pongor.vince@semmelweis.hu (V.F.-P.); csik.boglarka@semmelweis.hu (B.C.); mozes.noemi@semmelweis.hu (N.M.); trombitasne.feher.agnes@semmelweis.hu (Á.F.); dosa.norbert@semmelweis.hu (N.D.); arva.dorottya@semmelweis.hu (D.Á.); partos.katalin@semmelweis.hu (K.P.); kaposvari.csilla@semmelweis.hu (C.K.); horvath.krisztian@semmelweis.hu (K.H.); varga.peter@semmelweis.hu (P.V.); 2Fodor Center for Prevention and Healthy Aging, Semmelweis University, 1085 Budapest, Hungary; 3Doctoral College, Health Sciences Division, Semmelweis University, 1088 Budapest, Hungary

**Keywords:** western dietary pattern, cognitive decline, neuroinflammation, mediterranean diet, polyphenols, cerebrovascular aging, cognitive frailty, semmelweis study, workplace health promotion

## Abstract

Background: Accelerated demographic aging in Hungary and across Europe presents significant public health and socioeconomic challenges, particularly in preserving cognitive function and preventing neurodegenerative diseases. Modifiable lifestyle factors—especially dietary habits—play a critical role in brain aging and cognitive decline. Objective: This narrative review explores the mechanisms by which Western dietary patterns contribute to cognitive impairment and neurovascular aging, with specific attention to their relevance in the Hungarian context. It also outlines the rationale and design of the Semmelweis Study and its workplace-based health promotion program targeting lifestyle-related risk factors. Methods: A review of peer-reviewed literature was conducted focusing on Western diet, cognitive decline, cerebrovascular health, and dietary interventions. Emphasis was placed on mechanistic pathways involving systemic inflammation, oxidative stress, endothelial dysfunction, and decreased neurotrophic support. Key findings: Western dietary patterns—characterized by high intakes of saturated fats, refined sugars, ultra-processed foods, and linoleic acid—are associated with elevated levels of 4-hydroxynonenal (4-HNE), a lipid peroxidation product linked to neuronal injury and accelerated cognitive aging. In contrast, adherence to Mediterranean dietary patterns—particularly those rich in polyphenols from extra virgin olive oil and moderate red wine consumption—supports neurovascular integrity and promotes brain-derived neurotrophic factor (BDNF) and nerve growth factor (NGF) activity. The concept of “cognitive frailty” is introduced as a modifiable, intermediate state between healthy aging and dementia. Application: The Semmelweis Study is a prospective cohort study involving employees of Semmelweis University aged ≥25 years, collecting longitudinal data on dietary, psychosocial, and metabolic determinants of aging. The Semmelweis–EUniWell Workplace Health Promotion Model translates these findings into practical interventions targeting diet, physical activity, and cardiovascular risk factors in the workplace setting. Conclusions: Improving our understanding of the diet–brain health relationship through population-specific longitudinal research is crucial for developing culturally tailored preventive strategies. The Semmelweis Study offers a scalable, evidence-based model for reducing cognitive decline and supporting healthy aging across diverse populations.

## 1. Introduction

Europe is undergoing a demographic transformation, with its population aging at an unprecedented rate [1]. This demographic shift poses significant challenges to healthcare systems, social services, and economies across the continent [2]. In Hungary, as in many other European countries, the aging population presents a complex spectrum of health-related concerns, including an increased burden of age-related diseases and a growing prevalence of cognitive decline [3,4,5]. Understanding the determinants of unhealthy aging is of paramount importance in this context, offering a pathway to develop targeted interventions and policies aimed at promoting healthy aging and preserving cognitive function.

Hungary, like its European counterparts, faces the multifaceted consequences of unhealthy population aging [6,7,8]. The number of people aged 65 years and over is currently 1.8 million and rising, of whom 1.3 million have some form of disability [9]. Nearly 40% of adults reported that they had at least one chronic disease, which is higher than the European Union (EU) average [10]. Because of the prevalence of chronic diseases, 22% of people aged 65–75 years and 40% of people aged 75+ years considered their health to be poor or very poor [11]. Excessive bodyweight is a significant problem in Hungary, as the data indicate that two out of three Hungarian adults are either overweight or obese [12,13]. Many older adults in Hungary live with a high risk of metabolic syndrome, which contributes to a higher risk of cardiovascular disease, type 2 diabetes mellitus (T2DM), and other diseases [14]. It is also regrettable that Hungary has one of the highest cancer mortality rates in the world [15,16]. The escalating morbidity and mortality rates from age-related diseases underscore the urgent need for comprehensive strategies to address the challenges of unhealthy aging. Among these challenges, cognitive decline represents a critical issue, with implications for individual autonomy, healthcare expenditure, and societal well-being. To address these concerns effectively, it is imperative to unravel the complex interplay of factors contributing to unhealthy aging, with a particular focus on modifiable risk factors, such as dietary habits [17,18,19,20].

In this review, we aim to elucidate the role of unhealthy dietary habits, notably the Western diet [21,22,23,24,25,26], in contributing to age-related cognitive decline. By examining the mechanisms through which dietary patterns influence cognitive function and outlining the consequences of prolonged exposure to a Westernized dietary landscape, we seek to provide insights that inform preventive strategies, public health policies, and clinical interventions aimed at promoting healthy aging and preserving cognitive function in Hungary and beyond. Furthermore, this review aims to inform the design of the Semmelweis Study [27], a comprehensive investigation into the determinants of unhealthy aging in Hungary. By integrating insights from this review into the framework of the Semmelweis Study, we aspire to advance our understanding of the factors contributing to cognitive decline and inform evidence-based interventions to promote healthy aging and cognitive well-being among Hungarians.

### Methods

This review employed a narrative literature review methodology to synthesize current evidence on the relationship between dietary patterns—particularly the Western diet—and cognitive aging. Peer-reviewed scientific articles were identified through comprehensive searches in databases such as PubMed, Scopus, and Web of Science, without restrictions on publication year. Keywords included combinations of “Western diet,” “cognitive decline,” “cerebrovascular aging,” “neurodegeneration,” “Mediterranean diet,” “neuroinflammation,” “brain-derived neurotrophic factor (BDNF)”, and “4-hydroxynonenal (4-HNE)”. Selection criteria focused on studies that explored the mechanistic pathways linking dietary habits to neurobiological and vascular changes, including chronic inflammation, oxidative stress, insulin resistance, endothelial dysfunction, and alterations in neurotrophic signaling.

Inclusion criteria encompassed clinical trials, observational cohort studies, meta-analyses, and mechanistic studies in both human and animal models. Particular emphasis was given to evidence relevant to Central and Eastern European populations, with a focus on modifiable risk factors and culturally applicable intervention strategies. Additionally, studies that informed the rationale and design of the Semmelweis Study and the associated Semmelweis–EUniWell Workplace Health Promotion Model were highlighted.

## 2. Western Diet and Its Health Effects

### 2.1. Overview of the Western Diet

The Western diet is characterized by its reliance on processed foods, saturated fats, and refined sugars, while being low in fruits, vegetables, and whole grains [25,26]. This dietary pattern has become emblematic of modern eating habits in Western societies and is increasingly prevalent worldwide [28]. At the core of the Western diet are highly processed and convenience foods, which are often high in calories, sodium, and unhealthy fats [28,29]. These include fast food items, like burgers, fries, pizzas, and sugary snacks and beverages. The Western diet tends to prioritize taste, convenience, and shelf-life over nutritional quality, leading to a diet rich in energy-dense but nutrient-poor foods [29]. Key features of the Western diet include the overconsumption of red and processed meats, such as beef, pork, bacon, and sausage, which are often high in saturated fats and cholesterol [30,31,32,33,34,35,36,37,38,39,40,41]. These animal-derived fats contribute to elevated levels of low-density lipoprotein (LDL) cholesterol and are associated with an increased risk of cardiovascular diseases [30,31,32,33,34,35,36,37,38,39,40].

Additionally, the Western diet is characterized by a low intake of fruits and vegetables, which are rich sources of essential vitamins, minerals, fiber, and antioxidants [42]. The insufficient consumption of these nutrient-dense foods deprives the body of important micronutrients and phytochemicals that are crucial for maintaining optimal health and preventing chronic diseases [43,44,45]. Moreover, the Western diet is notorious for its high consumption of refined sugars and carbohydrates, including sugary drinks, candies, pastries, and white bread [28]. These sources of refined carbohydrates contribute to rapid increases in blood sugar levels, leading to insulin resistance, weight gain, and an increased risk of type 2 diabetes [46,47].

Overall, the Western diet is associated with a plethora of adverse health outcomes, including obesity, type 2 diabetes, cardiovascular diseases, hypertension, and metabolic syndrome [28,48]. Importantly, emerging evidence suggests that the Western diet may also have detrimental effects on cognitive health, contributing to the development and progression of age-related cognitive decline and neurodegenerative diseases, like Alzheimer’s disease (AD) [49,50,51].

Understanding the components and consequences of the Western diet is crucial for developing targeted interventions and public health policies aimed at promoting healthier dietary patterns and preserving cognitive function. Figure 1 illustrates the potential biological mechanisms linking components of the Western diet to neurodegeneration and cognitive decline.

### 2.2. Similarities Between the Hungarian Diet and the Western Diet

Unhealthy dietary patterns among the Hungarian population reflect many characteristics of the typical Western diet, as highlighted by the 2019 National Nutrition and Nutritional Status Survey (OTÁP 2019) [52]. According to this survey, 77% of Hungarian men and 60% of women are either overweight or obese, with these trends being particularly pronounced among older adults, individuals with chronic diseases, and those with lower educational attainment. Abdominal obesity, affecting approximately every second adult, represents a concerning marker for elevated risks of cardiovascular and cerebrovascular diseases [53].

The prevalence of obesity in Hungary ranks among the highest in the European Union [54]. Data from the Central Statistical Office (Központi Statisztikai Hivatal, KSH, 2022) Sustainable Development Goals indicator system show that 25% of the adult population is classified as obese (Body Mass Index, BMI ≥ 30), which is significantly higher than the European Union average (17%), with only Malta reporting worse figures. Additionally, the Organisation for Economic Co-operation and Development (OECD) 2023 Health at a Glance report states that Hungary’s obesity prevalence is 33.2%, exceeding the OECD average of 25.7% [8,55]. These findings further underscore the significant public health issue posed by obesity in Hungary. The combined prevalence of overweight (BMI ≥ 25) and obesity among adults reaches 65–70%, posing considerable public health challenges, particularly concerning cardiometabolic and neurodegenerative diseases [8,55].

Inadequate dietary habits substantially contribute to the persistence of these adverse health outcomes [56]. The KSH’s nationally representative We Can Act for Our Health survey (2019) revealed that only 55% of the population consumes fruit daily, while 9.4% rarely or never do so. Vegetable consumption is similarly suboptimal, with 45% eating vegetables daily, 46% several times per week, and 8.8% rarely or never [8,56]. Current recommendations advise a minimum intake of five servings of fruits and vegetables per day; however, only approximately 8% of adults meet this guideline [57]. These data underscore the urgent need for preventive interventions focused on improving dietary behaviors and enhancing health literacy. Accordingly, one of the primary objectives of the Semmelweis Study presented here is the long-term monitoring and improvement of these structural and lifestyle factors, which are essential for reducing public health risks and mitigating the incidence of chronic diseases.

Traditional Hungarian cuisine, deeply rooted in Hungarian culture, shares several characteristics with the Western diet [7]. Notably, the prevalent use of pork is a hallmark feature of Hungarian dishes. Noteworthy traditional dishes include stew, goulash, stuffed cabbage, and various other pork-based creations. Additionally, this culinary tradition embraces the generous use of fat-rich sour cream and heavy cream. Carbohydrate-rich side dishes are also common in Hungarian culinary tradition [58,59]. However, these culinary characteristics contribute significantly to an increased risk of obesity and related chronic diseases. The consumption of excessive sugars and refined flour in traditional Hungarian desserts reflects a fondness for sweet indulgences, posing metabolic concerns and cognitive health risks [60,61]. Additionally, white bread is prevalent in the Hungarian diet, contributing to low fiber intake and rapid spikes in blood sugar levels [62]. Limited consumption of seeds further exacerbates this imbalance in nutrient intake. Incorporating whole grain alternatives and a variety of seeds can help improve the nutritional profile of the diet and support overall health. Moreover, the high intake of saturated fats from pork and other animal sources, along with the low consumption of fruits and vegetables, mirrors patterns observed in the Western diet.

The Westernized Hungarian diet incorporates processed and unhealthy fats, such as fried fats and smoked meats [63]. Fat intake among Hungarians significantly exceeds the recommended threshold, with a substantial portion derived from animal fats, like pork [64]. This dietary pattern is mirrored in cholesterol intake, which consistently surpasses the recommended daily limit of 300 mg [12]. Various studies indicate that Hungarians consume approximately twice as much fat, particularly saturated fats compared to mono- and polyunsaturated fats, as well as cholesterol [65,66,67,68].

Despite Hungary’s ample sweet water resources ideal for fisheries, the Westernized Hungarian diet tends to exclude fish, missing out on its high protein content, favorable fatty acid composition, and potential health benefits. This is an unexpected phenomenon considering that fish could be an affordable source of nutrition, even for lower-income populations [69].

Additionally, an insufficient inclusion of raw vegetables leads to a deficiency of essential vitamins, minerals, and antioxidants crucial for healthy aging. Many Hungarian dishes consist of overcooked vegetables. A characteristic dish of Hungarian cuisine is a vegetable stew thickened with roux in traditional preparation [58]. The roux is made from melted lard, or other fats, and an equal amount of fine flour. The flour is toasted in the heated fat according to the nature of the dish, then removed from the heat, diluted with cold water, and stirred until smooth [70]. This emulsion is poured over the pre-cooked vegetables in seasoned water [70]. The lack of variety in vegetable consumption is another notable drawback, translating to missed opportunities to benefit from the unique nutrients and health advantages offered by different vegetables. Healthy vegetables typically overlooked in Hungarian cuisine include kale, asparagus, and artichokes. Incorporating a wider variety of vegetables can significantly enhance the nutritional quality of the diet and contribute to healthy aging [59,71,72,73]. Furthermore, the limited consumption of mushrooms, known for their nutrient-rich profile and cognitive benefits, represents a missed opportunity to enhance dietary quality [69,74].

Low fruit consumption is another concerning aspect of the Westernized Hungarian diet [75,76]. Despite the abundance of fruits rich in essential vitamins, fiber, and antioxidants, their incorporation into daily dietary habits remains limited, depriving individuals of potential health benefits [77].

Milk and dairy products constitute integral components of a healthy, well-rounded diet. In Hungary, despite a recent upturn in milk consumption, the average individual still falls short of the recommended values, consuming approximately 60 L of milk and 20 kg of dairy products annually, a figure below the European Union average.

Hungarian dietary habits are characterized by excessive salt intake [76,78], with reports indicating that it can be three to four times the recommended amount. This heightened salt consumption poses a substantial health risk, particularly in relation to hypertension and associated cardiovascular diseases [79].

Alcoholic beverages, particularly Hungarian wines and brandy, hold a special place in Hungarian culture. However, this fondness for alcohol has resulted in alarmingly high alcohol consumption rates in the country, with reports of excessive energy intake from alcohol, alcoholism, and associated health risks [80]. On average, each person in Hungary consumes about 10.4 L of pure alcohol [81,82], and more than a third of Hungarian men (36.9%) report unhealthy alcohol consumption patterns in comparison to the 7.2% of women [83,84]. Recent research in Hungary indicates that among adults aged 18–64 years olds, beer is the most consumed beverage, followed by wine, and then spirits [82]. Men consume a higher proportion, frequency, and quantity of all types of alcoholic beverages [85]. The number of heavy drinkers has increased, with nearly 10% of Hungary’s population being severely alcohol-dependent, compared to 3.7% in the European Union [86]. Alcohol significantly increases the risk of cardiovascular diseases, liver diseases, and many other age-related diseases [87,88,89,90]. Additionally, it accelerates the aging of the brain and reduces the mass of gray and white matter. Few are aware that one gram of alcohol contains almost twice as many calories (7 kcal/g) as one gram of sugar (4 kcal/g); its energy content is higher than proteins and carbohydrates, approximately equivalent to fats [88]. Moreover, alcohol is considered to have “empty” calories, as it lacks real nutritional value; our body cannot utilize or store it, and it has unfavorable effects on metabolism, liver, and inhibits protein synthesis [87,88].

In conclusion, the Westernized Hungarian diet shares several unfavorable characteristics with the typical Western diet, including highly processed foods, added sugars, unhealthy fats, and low nutrient density. Furthermore, it presents distinct challenges, including the consumption of smoked meats, excessive sugars and flour, limited fish intake, insufficient raw vegetables, and a deficit in fruit consumption. Addressing these dietary patterns is crucial for promoting cognitive well-being within the Hungarian population.

### 2.3. Role of Western Diet in Unhealthy Aging

The consumption of a Western diet has been strongly associated with accelerated and unhealthy aging processes, contributing to a spectrum of adverse health outcomes that negatively impact overall longevity and quality of life [29], whereas the consumption of healthy diets, including the Mediterranean diet, are casually linked to healthy aging trajectories [91,92,93,94,95,96,97,98,99,100,101,102,103,104,105,106,107,108,109,110,111]. Biological aging, defined by the progressive decline in physiological function and cellular integrity, is notably exacerbated by the detrimental effects of a Western dietary pattern [29,112].

One of the hallmark consequences of consuming unhealthy diets [113,114,115,116,117,118] is an accelerated biological aging trajectory, characterized by an increased risk of age-related diseases and mortality. Epidemiological studies have consistently shown that individuals adhering to a Western dietary pattern exhibit an older biological age, as reflected by markers of cellular aging, such as increased epigenetic age [119,120] and increased oxidative stress [113,114,115,116,117,118]. Moreover, the consumption of high levels of saturated fats, refined sugars, and processed meats—core components inherent to the Western diet—contributes to chronic inflammation and oxidative damage, further accelerating the aging process at the cellular level [121,122,123].

Unhealthy aging, characterized by a greater susceptibility to age-related diseases and functional decline, is also intensified by the consumption of a Western diet [124,125,126]. Individuals following a Western dietary pattern are at increased risk of developing chronic conditions, such as obesity, type 2 diabetes, cardiovascular diseases, hypertension, and metabolic syndrome, all of which are closely linked to accelerated aging and a reduced lifespan [127,128,129]. Furthermore, the low intake of nutrient-dense foods, such as fruits, vegetables, and whole grains in the Western diet, deprives the body of essential vitamins, minerals, and antioxidants, exacerbating age-related decline in physiological function and increasing vulnerability to disease [130].

Importantly, the detrimental effects of the Western diet extend beyond physical health to include cognitive function and brain aging [21,22,23,25]. Emerging evidence suggests that the consumption of a Western diet is associated with an increased risk of cognitive decline; neurodegenerative diseases, such as Alzheimer’s disease; and impaired cognitive function in later life [21,22,23,25]. The neuroinflammatory and oxidative stress pathways activated by the Western diet contribute to the accumulation of amyloid-beta plaques and tau tangles in the brain, resulting in neuronal dysfunction and cognitive impairment [131].

In summary, the consumption of a Western diet is intricately linked to accelerated and unhealthy aging processes, manifesting as a higher biological age, increased mortality from age-related diseases, and cognitive decline. Understanding the role of the Western diet in promoting unhealthy aging is essential for developing targeted interventions and public health strategies aimed at promoting healthier dietary patterns and preserving cognitive function across the lifespan.

## 3. Western Diet and Cognitive Health

### 3.1. Cognitive Decline and Dementia as a Global Health Issue

Cognitive decline and dementia represent urgent global health concerns, impacting individuals, families, and societies worldwide [132]. With aging populations and increasing life expectancy, the prevalence of cognitive impairment and dementia is on the rise, posing significant challenges for healthcare systems and economies [132].

The prevalence of cognitive decline and dementia varies across regions, but it is universally acknowledged as a major public health issue. Dementia affects approximately 15% of the population over sixty-five years old, around 20% of those over seventy-five years, and 30% or more in the age group over eighty-five years [133]. Globally, nearly 55 million people suffer from dementia, with estimates suggesting that this number could more than double by 2050 [134,135]. In Hungary, as in many other European countries, the prevalence of dementia is increasing due to demographic shifts toward an aging population. According to recent estimates, approximately 6.2% of Hungarians over the age of 60 years are living with dementia, corresponding to around 149,000 people. This figure is expected to rise further in the coming years as the population continues to age [136,137].

The burden of dementia extends beyond the affected individuals, profoundly affecting their families, caregivers, and communities [138]. From a societal perspective, the economic burden of dementia is substantial. The cost of caring for individuals with dementia, including medical expenses, long-term care, and lost productivity, is immense [139]. Dementia care consumes a significant portion of healthcare budgets in many countries, diverting resources from other critical areas of healthcare [140]. Beyond dementia, age-related cognitive impairment affects millions of individuals, manifesting as subtle declines in memory, attention, and executive function [141]. Age-related cognitive impairment can have profound effects on individuals, impacting their ability to perform daily tasks, maintain independence, and participate fully in social and economic activities [142]. Individuals with cognitive impairment often experience a decline in cognitive functions, such as memory, reasoning, and decision-making, affecting their ability to perform daily tasks independently [143]. This loss of autonomy can lead to increased dependency on caregivers and, in severe cases, necessitate institutionalization, placing a considerable emotional and financial strain on families. Even mild cognitive deficits can hinder productivity, diminish quality of life, and increase the risk of accidents and injuries [144]. From a societal perspective, age-related cognitive impairment poses challenges to workforce productivity, healthcare systems, and social support networks [145]. As the population ages, the burden of cognitive impairment is expected to grow, placing additional strain on already stretched healthcare resources and exacerbating socioeconomic disparities [146,147].

An increasingly recognized concept in the field of aging research is cognitive frailty, which refers to a clinical condition characterized by the simultaneous presence of physical frailty and cognitive impairment in the absence of overt dementia [148,149,150,151]. Cognitive frailty is thought to represent an intermediate, potentially reversible state of vulnerability that precedes more severe neurocognitive disorders [152]. Individuals with cognitive frailty are at increased risk of progression to dementia, disability, hospitalization, and mortality. Importantly, cognitive frailty is modifiable through lifestyle interventions—particularly through diet, physical activity, and psychosocial engagement [153]. Poor nutrition, physical inactivity, and cardiometabolic dysfunctions, all of which are linked to Western dietary patterns, are known contributors to both physical and cognitive frailty [154]. Thus, understanding and mitigating cognitive frailty through public health strategies may offer a valuable window of intervention to delay or even prevent the onset of dementia.

Prevention plays a crucial role in addressing the challenges posed by age-related cognitive impairment and dementia [155]. By investing in research, public awareness campaigns, and age-friendly policies and environments, we can work toward promoting cognitive health and enhancing the well-being of aging populations worldwide. Public health interventions, including dietary interventions aimed at promoting healthier eating habits, can play a vital role in reducing the risk of cognitive decline and dementia [156]. By adopting a holistic approach to prevention, we can empower individuals to take proactive steps to maintain cognitive health and preserve their overall well-being as they age [157,158].

### 3.2. Risk Factors for Cognitive Decline and Dementia

Understanding the risk factors associated with cognitive decline and dementia is crucial for developing effective prevention strategies. While age is the most significant risk factor for cognitive impairment and dementia, several other factors contribute to the onset and progression of these conditions [159]. Identifying and addressing these risk factors can help reduce the burden of cognitive decline and dementia on individuals and society.

Cardiovascular risk factors, such as hypertension [160,161,162,163], diabetes [164,165,166,167], obesity [168,169,170,171,172,173,174,175,176,177,178], and high cholesterol levels [179], have been consistently linked to an increased risk of cognitive decline and dementia. These risk factors contribute to the development of vascular cognitive impairment (VCID), promoting small vessel disease and atherosclerosis, which can impair blood flow to the brain, disrupt the blood–brain barrier (BBB), increase vascular fragility, and lead to cognitive impairment [180,181,182]. Managing cardiovascular risk factors [183,184] through lifestyle modifications, such as maintaining a healthy diet, regular exercise, and controlling blood pressure and cholesterol levels, is essential for preserving cognitive function [26,156,185,186].

Unhealthy lifestyle habits [187], including smoking, excessive alcohol consumption, and sedentary behavior [188,189], are associated with an elevated risk of cognitive decline and dementia [190,191]. Smoking and excessive alcohol consumption can damage blood vessels and increase inflammation in the brain, contributing to cognitive impairment [192]. Sedentary behavior and a lack of physical activity have also been linked to accelerated biological aging [193,194] and a higher risk of cognitive decline [195,196]. Adopting a healthy lifestyle [187] that includes regular exercise [197,198,199,200,201,202,203], weight management [204], healthy sleep habits [205], avoiding smoking and excessive alcohol consumption, and staying mentally and socially active can help reduce the risk of cognitive decline and dementia [206].

Dietary factors play a significant role in cognitive health, with mounting evidence suggesting that certain dietary patterns are associated with a lower risk of cognitive decline and dementia [207,208,209]. The Mediterranean diet, characterized by a high intake of fruits, vegetables, whole grains, fish, and healthy fats, has been consistently linked to better cognitive function and a reduced risk of dementia (Table 1) [94,210]. In contrast, diets high in saturated fats, refined sugars, and processed foods, such as the Western diet, are associated with an increased risk of cognitive impairment [211,212]. Promoting healthier dietary habits, including the adoption of Mediterranean-style eating patterns, can help protect against cognitive decline and dementia [94,96]. Part of the effects of diets are mediated through changes in the microbiome [213,214,215,216,217] and consequential alterations in the metabolome [218,219].

While genetic factors play a role in the development of cognitive decline and dementia, they interact with environmental and lifestyle factors to influence individual risk [156,220]. Certain genetic variants, such as the Apolipoprotein E (APOE) ε4 allele [221,222,223,224,225], have been associated with an increased risk of Alzheimer’s disease, the most common form of dementia. However, not everyone with these genetic risk factors will develop dementia, highlighting the importance of environmental factors in modulating risk [222,226,227]. Environmental factors, such as air pollution [228,229], occupational exposures [230], education level [231], socioeconomic status [232,233,234,235,236], and access to healthcare [6], also play a significant role in determining cognitive health outcomes. Engaging in intellectually stimulating activities [237,238,239,240], such as reading, puzzles, and lifelong learning, has been shown to have a protective effect against cognitive decline and dementia [241]. Similarly, maintaining social connections [242,243] and participating in social activities [244] and their beneficial effects on life satisfaction [245,246] can help preserve cognitive function in older adults. Cognitive and social engagement promote brain plasticity and resilience, helping to build cognitive reserve that may buffer against the effects of aging and neurodegenerative diseases [247]. Certain medical conditions, such as depression [248,249,250,251], sleep disorders [252], and untreated hearing loss [253], are associated with an increased risk of cognitive decline and dementia [254,255,256,257,258,259,260,261,262,263,264].

### 3.3. The Western Diet and Cognitive Decline: Preclinical and Clinical Evidence

The Western diet, characterized by its high consumption of processed foods, saturated fats, refined sugars, and a low intake of fruits, vegetables, and whole grains, has garnered increasing attention for its potential impact on cognitive health [211,265]. A growing body of research has explored the association between adherence to a Western dietary pattern and the risk of cognitive decline and dementia [29,266,267]. In this section, we delve into the evidence linking the Western diet to cognitive impairment and dementia, shedding light on the mechanisms underlying this relationship and its implications for public health.

#### 3.3.1. Epidemiological Studies

Numerous epidemiological studies have examined the association between dietary patterns resembling the Western diet and cognitive outcomes [268,269,270]. These studies have consistently found that individuals with a higher adherence to a Western dietary pattern are at an increased risk of cognitive decline, Alzheimer’s disease, and other forms of dementia [267,271]. Longitudinal cohort studies have demonstrated a dose–response relationship, with greater adherence to the Western diet correlating with a higher risk of cognitive impairment over time [272,273,274]. These findings underscore the importance of dietary factors in shaping cognitive health outcomes [275].

#### 3.3.2. Preclinical Studies and Experimental Evidence

Animal studies and experimental research on cellular and animal models [212,276] have provided further insights into the mechanisms through which the Western diet affects cognitive function [277,278,279]. These studies have demonstrated that a diet high in saturated fats and refined sugars [280] can induce blood–brain barrier disruption, thereby promoting neuroinflammation [281,282] and oxidative stress [283], microvascular rarefaction and dysfunction [284], dysregulation of cerebral blood flow, synaptic dysfunction [285], and neuronal damage in the brain, contributing to cognitive deficits. Moreover, dietary interventions aimed at mimicking the components of the Western diet have been shown to impair learning and memory performance in animal models, highlighting the detrimental effects of this dietary pattern on brain function. In contrast, dietary interventions aimed at activating anti-geronic cellular processes, including caloric restriction [20,286,287,288,289,290], ketogenic diets [291,292,293,294], and intermittent fasting regimens [295,296], have been shown to exert beneficial cognitive effects [297,298,299,300] associated with systemic anti-aging effects [301].

#### 3.3.3. Human Intervention Studies

While observational and experimental evidence suggests a link between the Western diet and cognitive decline, human intervention studies evaluating the effects of dietary interventions on cognitive outcomes are limited but promising [302]. Some intervention studies have shown that an adherence to a Mediterranean-style diet [94,96], characterized by a high consumption of fruits [303], vegetables, whole grains, fish [304,305], and healthy fats [94,96,305,306,307], can improve cognitive function and reduce the risk of stroke and dementia [21,24,94,96,99,100,101,109,110,111,305,308,309,310,311,312,313,314]. Conversely, interventions aimed at reducing the intake of saturated fats, refined sugars, and processed foods have shown potential benefits for cognitive health [315,316]. However, further research is needed to elucidate the long-term effects of dietary interventions on cognitive function and to identify optimal dietary patterns for promoting brain health.

The evidence linking the Western diet to cognitive decline has significant implications for public health policy and practice [317]. Promoting healthier dietary patterns, such as the Mediterranean diet, and discouraging the consumption of processed and unhealthy foods characteristic of the Western diet, could help reduce the risk of cognitive impairment and dementia at a population level [318]. Public health initiatives aimed at improving dietary habits, increasing access to nutritious foods, and raising awareness about the importance of diet for cognitive health are essential for addressing the growing burden of cognitive decline and dementia in aging populations [319]. By prioritizing preventive strategies and promoting brain-healthy lifestyles, policymakers, healthcare providers, and communities can take proactive steps to safeguard cognitive function and enhance the well-being of individuals as they age [320].

We now know that the so-called Western diet is extremely harmful, primarily due to its high intake of animal-derived saturated fats, trans-fatty acids, cholesterol, sugars, and salt [29]. It leads to elevated serum cholesterol levels, weight gain, diabetes, and hypertension, and an increase in coronary heart diseases [29]. Epidemiological data have shown that in countries where the consumption of animal fats has increased, coronary mortality has increased accordingly [321], while in countries where fat consumption has been reduced, coronary mortality has decreased [322,323]. The Lyon Heart Study [324] compared the effects of the Mediterranean diet and the Western diet, and showed a reduction in cardiovascular events by 65% due to the Mediterranean diet [324,325]. Fish consumption increases plasma omega-3 fatty acids levels, which play a crucial role in the prevention of arrhythmias, thromboses, and cardiovascular events [326]. Another study aimed to investigate whether diet influences the incidence of strokes. Patients were divided into three groups according to their vegetable and fruit consumption: Group 1 consisted of individuals who consumed vegetables and fruits less than three times a day, while Groups 2 and 3 consisted of individual who consumed them three to five times a day and individuals who consumed them more than five times a day. A total of 257,551 patients and 4917 strokes were analyzed over a 13-year follow-up period. It was found that, compared to the first group, the incidence of strokes significantly decreased in both the second and third groups, with an 11% reduction in the second group and a 26% reduction in the third group [325]. The beneficial effects of regular fruit and vegetable consumption are multifaceted: increased potassium intake reduces blood pressure, increased folic acid intake reduces homocysteine levels in the blood, and the antioxidant properties of fruits and vegetables are essential, inhibiting the oxidation of LDL cholesterol. All three mechanisms protect the integrity of the vascular system and prevent the development of atherosclerosis [327].

The characteristic features of the Western diet are well-known, including the consumption of foods fried in oil, red meat, foods rich in saturated fats, and excessive consumption of refined grains. Meanwhile, the dietary intake of fiber is very low, and a high sugar content (high glycemic index) leads to the early development of health problems. A study from 2017 [328] described how the consumption of unhealthy foods has an immediate impact on the hippocampus, even if consumed for a short period (in this study, it was four days), and the hippocampus is the part of the brain responsible for memory and regulating appetite. Another study [329] divided its participants into two groups: one group consumed a high-calorie Western diet for seven days, while healthy young individuals in the control group maintained their regular diet. In both groups, memory tests were conducted before and after meals, and those on the unhealthy diet performed significantly worse. The researchers pointed out that the decline in cognitive functions due to the Western diet becomes evident very quickly, affecting the functioning of the hippocampus, which may also play a role in appetite regulation. Furthermore, participants following a Western diet also tended to crave snacks and sweets when they were not hungry, immediately after finishing a main meal. This type of overeating often results in diabetes and obesity, all of which further increase the risk of developing dementia [330,331,332]. In summary, existing human intervention studies suggest that dietary modifications—particularly reducing components typical of the Western diet and adopting Mediterranean-style eating patterns—can have beneficial effects on cognitive function and may help prevent neurodegenerative diseases. While the current evidence is promising, further long-term, well-controlled studies are needed to better understand the impact of dietary interventions on brain health and to identify optimal nutritional strategies for preserving cognitive function across the lifespan.

#### 3.3.4. The Role of Extra Virgin Olive Oil and Moderate Red Wine Consumption in the Health-Promoting Effects of the Mediterranean Diet

Descriptions of the Mediterranean diet often emphasize the high consumption of vegetables, fruits, fish, and whole grains. However, two of its most characteristic and scientifically supported components—extra virgin olive oil (EVOO) and moderate red wine consumption—are surprisingly less frequently discussed, despite their substantial contributions to the diet’s health benefits.

EVOO is not only rich in monounsaturated fatty acids, but also contains significant amounts of bioactive polyphenolic compounds, such as hydroxytyrosol, tyrosol, oleuropein, and their secoiridoid derivatives (e.g., oleocanthal) [333]. These compounds have been extensively studied for their antioxidant, anti-inflammatory, anti-aging, and neuroprotective properties [334]. In vitro, animal, and human studies have confirmed their ability to inhibit lipid peroxidation, prevent LDL oxidation, reduce inflammatory markers such as C-reactive protein (CRP) and interleukin-6 (IL-6), and improve endothelial function [335,336]. The European Food Safety Authority has approved the following health claim regarding olive oil polyphenols: a daily intake of 5 mg of hydroxytyrosol, achievable through approximately 23 g of EVOO, protects blood lipids from oxidative stress [337].

Similarly, the moderate consumption of red wine, particularly when consumed with meals, may contribute to the beneficial effects of the Mediterranean diet—primarily through its resveratrol content [338]. Resveratrol is a stilbene-type polyphenol found in grape skins and extracted into wine during fermentation. It exhibits antioxidant, anti-inflammatory, cardioprotective, and neuroprotective effects, and may support insulin sensitivity, improve lipid metabolism, and help slow cognitive decline [339]. However, it is important to underscore that alcohol consumption also carries potential health risks, including liver damage and increased cancer risk [340]. Thus, the health-promoting effects of red wine can only be considered valid when consumed in moderation, with meals, and within an appropriate social and dietary context. In summary, the polyphenol content of extra virgin olive oil and the moderate intake of red wine represent two key features of the Mediterranean diet that contribute to its antioxidant, anti-inflammatory, and neuroprotective effects. Nonetheless, it must be emphasized that alcohol-related health risks should not be overlooked, and red wine should not be regarded as a standalone preventive agent, but rather as a potentially beneficial element when integrated mindfully into a balanced dietary pattern.

#### 3.3.5. The Mediterranean Diet and Its Association with Neurotrophic Factors

Recent research increasingly supports the notion that the Mediterranean diet is not only beneficial from an anti-inflammatory, antioxidant, and metabolic perspective, but also exerts direct effects on the expression of key neurotrophic factors—primarily brain-derived neurotrophic factor (BDNF) and nerve growth factor (NGF) [341]. These factors play crucial roles in synaptic plasticity, neurogenesis, and neuronal survival, and their reduced levels have been associated with neurodegenerative diseases, such as Alzheimer’s disease and depression [342,343].

A central element of the Mediterranean dietary pattern is the regular consumption of fresh vegetables combined with extra virgin olive oil, which provides a rich source of bioactive compounds, including polyphenols (e.g., oleuropein, hydroxytyrosol) and monounsaturated fatty acids [333]. Both animal and human studies have demonstrated that diets rich in EVOO enhance hippocampal BDNF expression, reduce oxidative stress, and inhibit apoptosis in brain tissues [344]. Collectively, these effects may contribute to the deceleration of cognitive decline and the reduction in neurodegenerative disease risk, particularly Alzheimer’s disease [344].

Beyond polyphenols, the Mediterranean diet includes high amounts of dietary fiber, flavonoids, and omega-3 polyunsaturated fatty acids, found in foods such as fish, nuts, and leafy green vegetables. These nutrients also positively influence BDNF and NGF levels, supporting neuronal regeneration and long-term brain health [48]. Together, these dietary components may help explain the consistently observed neuroprotective effects of the Mediterranean diet in epidemiological and clinical studies across various populations.

### 3.4. Mechanisms Linking the Western Diet to Cognitive Decline

The mechanisms underlying the association between the Western diet and cognitive decline are multifaceted and complex. High intakes of saturated fats and cholesterol, common components of the Western diet, can lead to dyslipidemia and vascular dysfunction [29], impairing blood flow to the brain and promoting the accumulation of amyloid-beta plaques and tau protein tangles, which are hallmark features of Alzheimer’s disease [345]. Additionally, excessive consumption of refined sugars and carbohydrates can lead to insulin resistance, inflammation, and oxidative stress, further exacerbating neurodegenerative processes and cognitive decline [316,346]. The lack of essential nutrients, such as vitamins, minerals, and antioxidants, in the Western diet may also compromise brain health and resilience, making individuals more susceptible to cognitive impairment [29]. The Western diet’s detrimental impact on cognitive function involves complex mechanisms that affect various aspects of brain health [347]. Several key pathways have been identified through preclinical and clinical research, shedding light on how dietary habits influence cognitive decline.

#### 3.4.1. Oxidative Stress and Inflammation

One of the primary mechanisms linking the Western diet to cognitive decline involves oxidative stress and inflammation in the brain [22,211,348]. The high consumption of processed foods, saturated fats, and refined sugars leads to an imbalance between the production of reactive oxygen species (ROS) and the body’s antioxidant defenses [349]. This imbalance results in oxidative damage to neuronal cells, impairing their function and contributing to cognitive decline. Additionally, the Western diet promotes chronic low-grade inflammation, characterized by elevated levels of pro-inflammatory cytokines [350,351]. This inflammatory response further exacerbates neuronal damage and disrupts synaptic plasticity, ultimately compromising cognitive function.

#### 3.4.2. Accelerated Cerebromicrovascular Aging

Accelerated cerebromicrovascular aging is a hallmark of cognitive decline [352,353,354,355,356] and plays a pivotal role in the pathogenesis of both vascular cognitive impairment and Alzheimer’s disease [357]. The relationship between Western dietary patterns and premature aging of cerebral microcirculation has been the subject of increasing scientific interest. The Western diet has been shown to induce the pathological remodeling of the cerebral microvasculature, leading to impaired vasodilation, endothelial dysfunction, and capillary rarefaction [23,49,358,359].

The aging of cerebral microcirculation is driven by a combination of molecular and cellular processes, among which increased endothelial and perivascular cell senescence [360] and mitochondrial dysfunction [361] are key contributors. Senescent endothelial cells lose their ability to maintain vascular homeostasis [362] and adopt a pro-inflammatory, pro-thrombotic phenotype, known as the senescence-associated secretory phenotype (SASP) [363,364,365]. This chronic, sterile inflammation [366] exacerbates oxidative stress and impairs neurovascular coupling, promotes blood–brain barrier disruption [367], microvascular fragility, and the genesis of cerebral microhemorrhages [368]. Additionally, mitochondrial dysfunction—marked by reduced ATP production, elevated reactive oxygen species, and impaired mitochondrial dynamics—further compromises vascular cell viability and barrier integrity, leading to blood–brain barrier breakdown and neuroinflammation [369].

Western dietary components exacerbate these pathological processes [370]. Diets high in saturated fats and sugars accelerate endothelial senescence by promoting oxidative damage and impairing mitochondrial biogenesis [371]. A low intake of antioxidants and polyphenols further reduces the brain’s resilience to oxidative stress [372]. Obesity [373] and insulin resistance [374]—commonly associated with Western dietary patterns—create a systemic pro-inflammatory milieu that accelerates vascular aging and disrupts cerebral perfusion [375].

Moreover, the detrimental effects of the Western diet are not limited to the microvasculature. Large vessel atherosclerosis, another consequence of poor dietary habits, is intricately linked with small vessel disease (SVD) [376,377,378]. Atherosclerosis of the carotid and cerebral arteries can impair cerebral autoregulation [379], reduce downstream perfusion, exacerbate BBB disruption [380], and promote microvascular ischemia [381]. Studies have shown that markers of large artery stiffness and intima-media thickening are associated with the burden of cerebral white matter lesions [380,381,382], microbleeds [378], and cognitive decline. Thus, poor dietary habits contribute to a vicious cycle of macrovascular and microvascular dysfunction, compounding the risk of dementia.

Understanding the mechanistic interplay between poor dietary habits, vascular aging, and cognitive impairment provides a strong rationale for public health interventions. Promoting anti-inflammatory, antioxidant-rich diets—such as the Mediterranean [91] or Dietary Approaches to Stop Hypertension (DASH) diets [383]—may help preserve cerebrovascular health by mitigating senescence, enhancing mitochondrial function, and preventing both large and small vessel pathology. From a preventive standpoint, these findings support dietary interventions as a viable strategy to reduce the burden of age-related cognitive disorders.

#### 3.4.3. Insulin Resistance and Metabolic Dysfunction

Another significant mechanism by which the Western diet affects cognitive health is through obesity [22,384,385], the development of insulin resistance, and metabolic dysfunction [386,387,388]. The excessive consumption of refined carbohydrates and sugars leads to chronically elevated blood glucose levels, triggering insulin resistance in peripheral tissues and the brain [389,390]. Insulin resistance impairs insulin signaling pathways in the brain, disrupting glucose uptake and metabolism in neuronal cells [391]. This metabolic dysfunction not only impairs energy production in the brain but also promotes the accumulation of toxic protein aggregates, such as amyloid-beta plaques, characteristic of Alzheimer’s disease [392]. Furthermore, insulin resistance contributes to vascular dysfunction and compromises cerebral blood flow, further exacerbating cognitive decline [393].

Insulin resistance is often associated with hyperglycemia, hyperinsulinemia, hypertension, dyslipidemia, and central obesity, collectively increasing the risk of developing metabolic syndrome [394]. Chronic hyperglycemia and hyperinsulinemia directly contribute to the overproduction of reactive oxygen species and the formation of advanced glycation end products (AGEs) [395]. Protein glycation and increased oxidative stress are the two main mechanisms in the biological aging process [396]. Both are likely associated with the etiopathogenesis of Alzheimer’s disease, with diabetes itself being a significant risk factor for the development of various dementias [396].

Diabetes mellitus is a common and significant chronic and systemic disease that can damage various organs in the body [397]. In addition to associated micro- and macrovascular complications, it can also cause alterations in the central nervous system, ultimately leading to cognitive decline [332]. Numerous studies demonstrate that individuals with obesity and/or abnormal insulin metabolism are more prone to the development of cognitive impairments and dementia [398,399,400]. This susceptibility extends to pre-diabetic conditions and metabolic syndrome as well [401]. Overall, the research findings indicate that the higher an individual’s blood sugar level, the faster the rate of cognitive decline [402,403]. Furthermore, glucose directly contributes to the shrinkage of the hippocampus, implying that excessive sugar consumption can impair memory, even in the absence of insulin resistance and/or established diabetes [22,404]. Strategies aimed at reducing glucose levels can positively influence cognition in the elderly population, emphasizing their importance in prevention, as anything promoting insulin resistance ultimately increases the risk of dementia [405]. In summary, diabetes mellitus can expedite the aging processes of the brain, reduce cognitive performance, and lead to symptoms of dementia.

#### 3.4.4. Dysbiosis of the Gut Microbiota and Cognitive Decline

Growing evidence suggests that the Western diet may contribute to cognitive decline not only directly, but also through alterations in the gut microbiota composition—referred to as dysbiosis [317,406,407,408]. The habitual consumption of highly processed, high-fat, and low-fiber foods typical of the Western diet disrupts the balance between beneficial and harmful gut bacteria, leading to reduced microbial diversity [409,410]. This dysbiotic state increases intestinal permeability and triggers systemic inflammation, which can negatively affect brain function through the microbiota–gut–brain axis [411]. Disrupted microbial communities produce metabolites that influence neurotransmitter synthesis, neuroinflammation, and neuronal function—mechanisms that may underlie cognitive impairments [412].

The gut–brain axis operates through a bidirectional communication system, in which gut bacteria can directly produce neurotransmitters and modulate the immune system, indirectly impacting behavior and cognition. Studies show that greater microbial diversity supports better memory, reduces anxiety, and enhances cognitive performance [413,414,415].

In obese individuals and those with chronic low-grade inflammation—both common consequences of the Western diet—alterations in the gut microbiota are commonly observed, including an increased *Firmicutes*-to-*Bacteroidetes* ratio and a higher relative abundance of *Actinobacteria* [416,417,418]. However, the relationship between *Bacteroidetes* and metabolic health is complex: some studies report a decrease in *Bacteroidetes* in obesity [419], while others have shown increased proportions of *Bacteroidetes* (particularly certain *Bacteroides* and *Prevotella* species) following dietary interventions that lead to weight loss [419]. These microbiota shifts have been linked to increased intestinal permeability and systemic inflammation, contributing to insulin resistance and a higher risk of cognitive decline. Importantly, the effects of diet-induced changes in the gut microbiota composition may vary depending on specific taxa and host factors, suggesting that the role of *Bacteroidetes* in inflammation and metabolic health cannot be generalized without considering context and microbial resolution at the genus or species level [420,421,422,423].

Altered gut microbiota has been implicated in the pathogenesis of several neurological disorders, including Alzheimer’s disease. Patients with AD often show reduced microbial diversity, along with a decreased abundance of beneficial bacteria, such as *Firmicutes* and *Bifidobacterium* [424,425,426]. However, the findings regarding *Bacteroidetes* are mixed: some studies report a reduction in *Bacteroidetes* in AD, while others show increased levels of specific pro-inflammatory strains within this phylum, such as *Bacteroides* species [427]. These microbiota alterations have been associated with heightened neuroinflammation, increased amyloid-β accumulation, oxidative stress, and exacerbated cognitive decline. Persistent dysbiosis may contribute to chronic low-grade systemic inflammation, a recognized factor in neurodegeneration [428,429,430].

In the long term, unhealthy Western-style diets may impair the function of the microbiota–gut–brain axis. Low fiber intake and excessive dietary fat reduce gut microbial diversity, which has been associated with poor mental health, accelerated cognitive decline, and a greater risk of neurodegenerative diseases [29,406]. The gut microbiota’s composition and metabolic output—shaped by diet—play a key role in regulating neuroinflammatory processes that influence cognitive outcomes. Therefore, modifying dietary habits and optimizing the gut microbiome through increased fiber intake, reduced consumption of processed foods, and the use of pro- and prebiotics may be essential strategies for maintaining cognitive health and preventing neurodegenerative diseases. The research in this field is ongoing, but the current findings underscore the importance of a healthy diet and balanced gut microbiota in supporting long-term cognitive function.

#### 3.4.5. Impaired Synaptic Plasticity and Neurotrophic Support

Furthermore, the Western diet’s influence on cognitive decline involves impaired synaptic plasticity and reduced neurotrophic support [211,347]. Essential nutrients found in fruits, vegetables, and whole grains play crucial roles in supporting neuronal growth, synaptic connectivity, and neurogenesis. However, the low intake of these nutrient-dense foods in the Western diet deprives the brain of the necessary building blocks for maintaining healthy neuronal function. Additionally, high levels of saturated fats and cholesterol disrupt membrane integrity and interfere with neurotransmitter release and receptor function, impairing synaptic transmission. The lack of neurotrophic factors, such as BDNF [176,431], further compromises neuronal survival and plasticity, contributing to cognitive decline.

#### 3.4.6. Role of the Consumption of Ultra-Processed Foods

Ultra-processed foods (UPFs) have become a dominant component of modern diets worldwide and are increasingly implicated in adverse health outcomes, including cognitive decline [432,433,434,435,436,437]. These products—often rich in synthetic additives, preservatives, emulsifiers, colorants, and refined carbohydrates—are designed to be highly appealing in taste and texture, thereby encouraging overconsumption while providing minimal nutritional value. Mechanistically, the excessive intake of certain components found in UPFs, such as omega-6 linoleic acid, has been shown to promote mitochondrial dysfunction, oxidative stress, and neuronal apoptosis, all of which accelerate neurodegenerative processes [438,439,440].

A typical Western diet contains omega-6 to omega-3 ratios that far exceed physiological needs, contributing to a pro-inflammatory state [441]. Although the Hungarian diet differs in some respects from the American pattern, it shares many of the same problematic features. For example, sunflower oil—a widely used cooking oil in Hungary—contains approximately 60% omega-6 linoleic acid, and its heavy use contributes to the imbalance between pro-inflammatory and anti-inflammatory fatty acids. While healthier alternatives, such as rapeseed and olive oil are available, they remain underutilized in Hungarian households and institutional kitchens.

The health consequences of UPFs are extensive and well documented [442]. Frequent consumption of these foods has been associated with obesity, insulin resistance, hypertension, dyslipidemia, and systemic inflammation [443]. These conditions not only elevate the risk of cardiometabolic disease, but are also major contributors to cerebrovascular aging and cognitive impairment [444]. Importantly, observational studies and clinical trials have demonstrated that high intake of UPFs correlates with poorer performance in memory and executive function tasks, as well as increased prevalence of mood disorders, such as depression and anxiety [438,439,445].

Experimental studies further support these findings. In a randomized crossover trial conducted by the U.S. National Institutes of Health (NIH), participants consumed diets matched for total calories, macronutrients, sugar, sodium, and fiber—but one diet was composed of 80% ultra-processed foods, while the other was entirely unprocessed. Within two weeks, participants on the UPF-rich diet gained weight, consumed more calories, and exhibited hormonal changes indicative of disrupted appetite regulation, including elevated ghrelin and reduced leptin levels. Notably, these effects occurred even in healthy, young individuals and reversed upon returning to an unprocessed diet [445].

The implications for Hungary are profound. With two-thirds of adults classified as overweight or obese and ultra-processed foods becoming increasingly prevalent in both urban and rural areas, the metabolic and cognitive burdens of dietary habits are escalating [446,447,448]. The traditional Hungarian diet—once centered around homemade meals—has increasingly incorporated convenience foods, store-bought baked goods, and ready-to-eat meat products, mirroring trends seen in Western countries [449]. These shifts are particularly problematic in populations with lower socioeconomic status, where cheaper, energy-dense processed foods often displace healthier options [435,445,450].

At the mechanistic level, one of the most concerning effects of UPFs is their disruption of the gut–brain axis [451]. UPFs reduce gut microbial diversity and promote intestinal dysbiosis, leading to increased intestinal permeability and systemic inflammation [410]. This chronic inflammatory state can exacerbate neuroinflammation and has been associated with the pathogenesis of Alzheimer’s disease and other forms of dementia [452]. Additionally, diminished microbial diversity impairs immune resilience, a relationship that was brought into focus during the COVID-19 pandemic [49,453,454,455].

Given the widespread consumption and accessibility of UPFs, reducing their intake presents a significant public health challenge. These foods are inexpensive, shelf-stable, and aggressively marketed, often making them the default choice in time-constrained environments, such as workplaces [442]. In Hungary, targeted interventions are urgently needed. These may include nutrition education programs, subsidies for fresh produce, the reform of institutional meal planning (e.g., in workplace cafeterias), and policy-level measures to limit the marketing and availability of ultra-processed foods. In summary, the high consumption of ultra-processed foods poses a growing threat to metabolic and cognitive health in Hungary. As part of a broader strategy for healthy aging, reducing UPF intake—alongside promoting nutrient-dense, minimally processed diets—should be a key focus of national dietary guidelines and health promotion programs, including those implemented through the Semmelweis Workplace Health Promotion Model Program.

#### 3.4.7. Deep-Frying with Linoleic Acid-Rich Oils and the Role of 4-Hydroxynonenal in Neurodegeneration

A characteristic feature of Western and “Westernized” Hungarian dietary patterns is the frequent use of deep-frying at high temperatures with vegetable oils rich in linoleic acid (an ω-6 polyunsaturated fatty acid), such as sunflower or corn oil [456]. This cooking method promotes the formation of lipid peroxidation products, notably 4-hydroxynonenal (4-HNE), a well-documented mediator of oxidative stress and inflammation [457]. 4-HNE can damage proteins, lipids, and nucleic acids at the cellular level, and has been shown to disrupt neuronal ion homeostasis, impair mitochondrial function, and induce apoptosis [458].

4-HNE appears to be particularly toxic to neural tissues [459]. It has been shown to enhance amyloid beta-mediated free radical production and contribute to synaptic dysfunction [460]. Furthermore, 4-HNE may destabilize the heat shock protein 70.1 (Hsp70.1), a key factor in protein recycling and lysosomal integrity [461]. The impairment of these processes may lead to progressive neuronal degeneration. Recent studies suggest that Hsp70.1 dysfunction induced by 4-HNE—rather than Aβ accumulation alone—may represent a central pathogenic mechanism in Alzheimer’s disease and other neurodegenerative disorders [462]. Additionally, reduced activity of aldehyde dehydrogenase 2, the enzyme responsible for 4-HNE detoxification, has been associated with increased neuronal cell death in ischemic and neurodegenerative contexts [458].

## 4. Implications for the Semmelweis Study and the Semmelweis Workplace Health Promotion Model Program

The Semmelweis University, as a leading educational and healthcare institution in Hungary, is uniquely positioned to advance health promotion research and implementation. Central to this endeavor are the Semmelweis Study [27] and the Semmelweis Workplace Health Promotion Model Program, both designed to enhance the health and well-being of university employees while contributing to broader public health efforts [27,463].

The Semmelweis Study is a prospective, longitudinal workplace cohort study initiated in 2022, targeting university employees aged 25 years and above. Its primary objective is to elucidate the determinants of unhealthy aging in Hungary, with a specific focus on age-related diseases, such as cardiometabolic and cognitive conditions. Participants are enrolled voluntarily and must meet inclusion criteria, including active employment, capacity to consent, and an absence of pre-existing neurodegenerative conditions. The estimated sample size (~1500 individuals) is informed by power calculations aimed at detecting small-to-moderate effect sizes in dietary and cognitive associations.

Data collection includes socio-demographic surveys, validated dietary assessments (e.g., Food Frequency Questionnaire), clinical measurements (e.g., anthropometry, blood pressure, blood chemistry), and biobanking for future biomarker analyses. Cognitive functions are assessed using standardized tools, such as the Montreal Cognitive Assessment (MoCA) and the Digit Symbol Substitution Test (DSST). The primary outcomes encompass cognitive performance and cerebrovascular markers, while secondary outcomes include mental health indicators and work-related well-being. Follow-up assessments are planned at five-year intervals, enabling longitudinal tracking of risk factor trajectories and health outcomes over time.

Multivariable linear and logistic regression models examine the associations between dietary patterns and outcomes, adjusting for potential confounders, such as age, socioeconomic status, physical activity, and comorbidities. Propensity score matching and stratified analyses are applied to address bias and strengthen validity.

Dietary factors are central to the study’s conceptual framework, given their significant role in shaping health behaviors and risk profiles within the Hungarian population. Poor nutrition—characterized by an excessive intake of saturated fats and simple carbohydrates, and insufficient fiber and antioxidants—contributes to chronic inflammation, oxidative stress, and insulin resistance, all of which are implicated in cognitive decline and cerebrovascular disease.

By assessing a wide array of lifestyle, socioeconomic, psychological, and environmental factors, and employing repeated measurements at five-year intervals, the study offers a unique opportunity to track the evolution of dietary behaviors and their long-term health consequences. It also provides a platform for developing culturally tailored interventions aimed at promoting healthier aging and mitigating cognitive decline.

Leveraging insights from the Semmelweis Study, the Semmelweis Workplace Health Promotion Model Program [96,246] endeavors to translate research discoveries into actionable interventions aimed at enhancing the physical, mental, and social well-being of university employees. With a specific focus on fostering healthy aging, the program aims to advance initiatives conducive to healthy aging, instill a culture of wellness among employees, and align with national public health objectives. Informed by the findings of the Semmelweis Study, the Semmelweis Workplace Health Promotion Model Program is poised to implement evidence-based interventions aimed at promoting healthy lifestyles and preventing age-related diseases. By addressing dietary factors, promoting physical activity, and controlling cardiovascular risk factors, the program aligns with broader efforts to enhance cognitive function, quality of life, and healthy aging. The Semmelweis Workplace Health Promotion Model Program aims to contribute to the National Healthy Aging Program by developing a model that can be replicated throughout Hungary through the Hungarian Health Development Offices and Health Promotion Hospitals, thus contributing to national public health and healthy aging goals.

Addressing the dietary challenges of Hungary requires a multifaceted approach to intervention planning. It is imperative to communicate the importance of high-quality, modern, and balanced nutrition to the entire Hungarian population. To promote healthier eating habits, numerous interventions and initiatives can be considered. Educational programs and campaigns can help people understand the principles of healthy eating, the importance of proper nutrition, and how to compose a balanced, healthy diet. Support for healthy food choices in restaurants, workplace cafeterias, and other dining venues can be facilitated by making high-fiber foods, fresh fruits and vegetables, lean meats, and whole grains more readily available. The overall economic policy of taxing unhealthy foods or supporting healthy ones can help promote the choice of healthy foods while making less healthy foods more expensive. Programs and incentives supporting healthy eating habits at workplaces, informed by the Semmelweis Workplace Health Promotion Model Program, can be introduced. Campaigns and initiatives conducted with the participation of communities and civil organizations can also be crucial in promoting healthy eating habits.

Aligned with the Hungarian dietary guidelines, several recommendations can facilitate the establishment and maintenance of healthier dietary practices conducive to healthy cerebrovascular and brain aging. Firstly, there should be a concerted effort to increase the consumption of vegetables and fruits, along with promoting locally available, seasonal produce. Secondly, emphasis should be placed on whole grains. Thirdly, promoting healthy protein sources, such as lean meats, fish, eggs, legumes, and nuts, is crucial for brain health. Reducing salt and sugar intake is another essential recommendation. Encouraging the use of herbs and spices in meal preparation, along with minimizing the consumption of sweeteners and high-sugar foods, can support cognitive function and reduce the risk of cerebrovascular diseases. Promotion of healthy snacks, like fruits, nuts [464], or yogurt, in workplaces is also vital for brain health. Providing nutritional education sessions and workshops at workplaces, along with opportunities for employees to seek nutritional counseling, can empower individuals to make informed dietary choices that support brain health. Lastly, promoting physical activity by encouraging regular exercise [465,466,467,468,469,470] and providing opportunities for using gym facilities at or near workplaces can complement efforts to improve dietary habits and support healthy cerebrovascular and brain aging. However, in order to ensure the real-world effectiveness of such national-level dietary interventions, it is crucial to consider the regional and cultural specificities of the Hungarian population.

## 5. Regional and Cultural Specificities in the Hungarian Population

In addition to the well-documented adverse effects of the Western diet in the international literature, it is essential to highlight specific national characteristics that may further amplify the risk of cognitive decline mediated by dietary factors. In Hungary, dietary patterns have been historically characterized by a predominance of high-fat, animal-based meals, low consumption of fruits and vegetables, and the growing prevalence of ultra-processed foods—particularly among lower-income groups [56,471,472]. The proportion of household expenditures on food remains higher than the EU average, while access to fresh, high-quality ingredients exhibits significant socioeconomic disparities.

Multiple nationally representative surveys (e.g., OTÁP, data from the Hungarian Central Statistical Office [KSH]) confirm that the intake of saturated fats and simple carbohydrates among Hungarian adults exceeds recommended levels, while fiber and antioxidant intake often falls below optimal thresholds [56,67,473]. These dietary patterns are consistent with the characteristics of the Western diet and may contribute to systemic inflammation, insulin resistance, and oxidative stress—all of which are known to play a role in the development of neurodegenerative processes [29].

Cultural factors also play a significant role. Traditional Hungarian cuisine often features energy- and fat-dense dishes that are deeply embedded in family and festive contexts [474]. In addition, low levels of physical activity and the prevalence of workplace stress further contribute to the emergence of unhealthy lifestyle patterns [54,475].

It is important to emphasize that the effectiveness of health promotion interventions depends largely on their ability to account for these local cultural and structural conditions. One of the central objectives of the Semmelweis–EUniWell Workplace Health Promotion Model Program is to develop culturally adapted, realistic, and sustainable interventions for the Hungarian working population. These interventions aim to reflect traditional dietary habits, address accessibility issues, and consider the motivational and social drivers of behavior change. In light of these considerations, it is essential that future research on the link between Western diets and cognitive decline in Hungary explicitly addresses the cultural and nutritional specificities of the Hungarian population and formulates contextually relevant prevention strategies based on these insights [94,96,246,385,448,476,477,478,479,480,481].

## 6. Conclusions

In conclusion, the investigation of dietary factors and their implications for healthy cerebrovascular and brain aging within the Hungarian context underscores the critical importance of adopting and promoting healthier eating habits. With an aging population and rising rates of age-related cognitive decline and dementia, there is an urgent need to address dietary patterns and their impact on brain health. By drawing upon insights from research studies, like the Semmelweis Study, and implementing evidence-based interventions, such as the Semmelweis Workplace Health Promotion Model Program, Hungary can take proactive steps toward promoting healthy cerebrovascular and brain aging. Encouraging the consumption of brain-healthy foods, like vegetables, fruits, whole grains, and healthy proteins, while reducing salt and sugar intake, can support cognitive function and reduce the risk of cerebrovascular diseases and cognitive decline. Moreover, providing nutritional education, promoting physical activity, and creating supportive workplace environments can empower individuals to make healthier choices and foster a culture of healthy aging.

However, despite the growing evidence linking dietary patterns to brain health, the strength and consistency of this association remain a subject of ongoing scientific debate. Several systematic reviews and meta-analyses have reported that while healthy dietary patterns, such as the Mediterranean or DASH diets, are associated with better cognitive outcomes, the overall effect sizes tend to be modest, and the findings are sometimes inconsistent across populations and methodologies. These discrepancies may stem from variations in study design, dietary assessment tools, cognitive outcome measures, and confounding factors, such as education, physical activity, and socioeconomic status.

Therefore, while promoting healthy dietary habits remains a promising strategy for supporting brain health, it is important to interpret the current evidence with caution and acknowledge the need for further high-quality, longitudinal studies—particularly in Central and Eastern European populations. Integrating dietary interventions into comprehensive lifestyle approaches, supported by policy and education, may offer the most realistic and sustainable path forward. Through collaborative efforts across sectors, Hungary can pave the way for healthier aging and improved cognitive well-being, ensuring a brighter future for generations to come.

## Figures and Tables

**Figure 1 nutrients-17-02446-f001:**
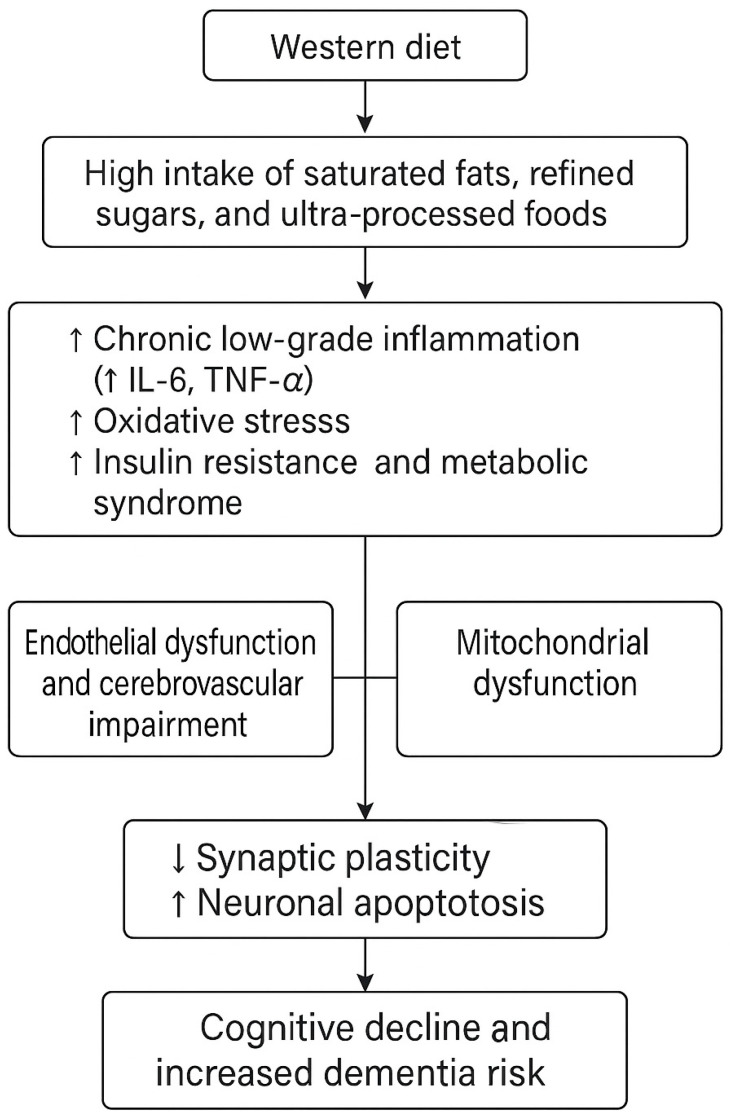
Mechanistic pathways linking the Western diet to cognitive decline. Arrows indicate an increase (↑) or decrease (↓) in processes.

**Table 1 nutrients-17-02446-t001:** Key dietary characteristics of the Mediterranean and Western dietary patterns.

Mediterranean Diet	Western Diet
Extra virgin olive oil	Refined vegetable oils and trans fats
Fish and seafood	Processed meats and excessive red meat consumption
High intake of vegetables and legumes	Low vegetable consumption
Whole grains	Refined grains and white flour
Moderate wine consumption (source of resveratrol)	Sugary beverages and excessive alcohol intake
Nuts and seeds	High-sugar snacks and desserts
Rich in antioxidants and polyphenols	Pro-inflammatory, low in essential micronutrients

This table was developed by the authors based on a synthesis of evidence from recent reviews on dietary patterns and their health implications.

## Data Availability

Data sharing is not applicable to this article as no new data were created or analyzed in this study.

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
