# Peer review of "Western Diet and Cognitive Decline: A Hungarian Perspective—Implications for the Design of the Semmelweis Study"

_nutrients, 2025, doi:10.3390/nu17152446_

Round 1
Reviewer 1 Report
Comments and Suggestions for Authors
This is a well written and comprehensive manuscript. It does a very good job of reviewing what is known about the relationship between diet and aging-related cognitive decline. It carefully describes research that has warned of a relationship between the Western diet and multiple aspects of aging and neurodegeneration. One question: you state that individuals with obesity and low grade inflammation exhibit decreased bacteroides, which promotes systemic inflammation. You then state that patients with AD show an increase in bacteroidetes, which you say is pro-inflammatory. Which is true?
Author Response
Response to Reviewer 1:
Thank you for pointing out the previously unclear and potentially misleading wording regarding the changes in Bacteroides and Bacteroidetes proportions in the manuscript. We fully agree that the original phrasing was ambiguous and did not adequately reflect the complexity of current scientific evidence.
We have revised the relevant section accordingly. The updated text provides a more accurate discussion of the Firmicutes/Bacteroidetes ratio in relation to obesity and highlights that while some studies report a reduction in Bacteroidetes, others—particularly those involving dietary interventions—show an increase in specific Bacteroidetes taxa following weight loss. We also clarified that such shifts in the gut microbiota are associated with increased intestinal permeability, sustained systemic inflammation, insulin resistance, and cognitive decline. Furthermore, we emphasized that the role of Bacteroidetes cannot be generalized, as the biological effects of individual taxa vary, and interpretation must take into account the microbial resolution level (genus, species) and contextual factors.
Your comment regarding Alzheimer’s disease-related microbiota changes was also well taken. We acknowledge that the previous version of the manuscript did not sufficiently capture the complex and sometimes contradictory nature of the current literature in this field. The revised text now highlights the frequently observed reduction in microbial diversity in AD patients, the decreased abundance of beneficial taxa such as Firmicutes and Bifidobacterium, and the increased presence of certain pro-inflammatory Bacteroides species. These alterations have been associated with enhanced neuroinflammation, amyloid-β accumulation, oxidative stress, and accelerated cognitive decline. We have also expanded the discussion to reflect the potential role of chronic low-grade systemic inflammation as a contributing factor in neurodegenerative processes.
Once again, thank you for your constructive comments and thoughtful suggestions,
with kind regards and sincere appreciation,
The Authors
Reviewer 2 Report
Comments and Suggestions for Authors
In this insightful review, the authors examine the rapid demographic aging of the Hungarian and broader European populations. This issue presents urgent public health and economic challenges, including the preservation of cognitive function and the prevention of age-related diseases. Indeed, emerging evidence highlights the role of modifiable lifestyle factors—particularly diet—in influencing brain aging and cognitive decline. The authors explore the adverse effects of the Western diet on cerebrovascular and cognitive health, emphasizing parallels with traditional Hungarian dietary patterns. The authors state that diets high in saturated fats, refined sugars, and ultra-processed foods—common features of both Western and Westernized Hungarian diets—contribute to systemic inflammation, oxidative stress, endothelial dysfunction, and metabolic disturbances that accelerate cerebral microvascular and neuronal aging. In contrast, healthier dietary patterns such as the Mediterranean diet are associated with reduced risk of cognitive decline, dementia, and cerebrovascular disease. The authors introduce the concept of cognitive frailty as a potentially reversible intermediate stage between normal aging and dementia, highlighting the preventive potential of dietary and lifestyle interventions. In this paper, the authors also show the rationale and design of the Semmelweis Study—a prospective cohort study of employees of Semmelweis University—and the Semmelweis-EUniWell Workplace Health Promotion Model Program, which together aim to assess and mitigate the dietary and lifestyle determinants of unhealthy aging. The authors concluded their work discussing how longitudinal research can inform effective public health strategies to promote healthy cognitive aging in Hungary and beyond.
The paper is potentially publishable; however, some modifications, in my opinion, could potentiate the paper.
-The English quality of the text could be improved.
-The Western/Hungarian dietary patterns are characterized by high utilization of deep-frying with linoleic acid-rich oils, a practice that elevates the formation of harmful lipid peroxidation products, notably 4-hydroxynonenal, known to be implicated in accelerating the aging process. This issue is not considered in the paper and could be mentioned/discussed.
-The paper mentions the Mediterranean diet in several parts of the text. However, oddly, in these sentences there is no presence of the regular consumption of extra-virgin olive oil known to contain polyphenols (Hydroxytyrosol, Tyrosol, Oleuropein) with antioxidant, antiaging, anti-inflammatory, and anti-neurodegenerative properties. Further, no mention in the text of the moderate consumption of red wine, containing resveratrol, a polyphenol known to possess actions comparable to those exerted by the EVO polyphenols.
-The authors could make more effort in discussing the tight relationship between neurotrophic factors (such as NGF and BDNF), known to play a crucial role in modulating neurodegeneration and the Mediterranean diet. Indeed, the regular consumption, in this dietary pattern, of fresh vegetables dressed with EVO could exert powerful anti-inflammatory, antioxidant, and anti-apoptotic activities.
Author Response
Dear Reviewer,
We would like to sincerely thank you for your thorough, constructive, and highly valuable feedback. We truly appreciate your recognition of the strengths of our manuscript while offering insightful and forward-looking suggestions for its improvement.
We fully agree that the use of linoleic acid-rich oils for deep-frying—and the resulting formation of 4-hydroxynonenal, especially given its role in accelerating aging and contributing to neurodegenerative processes—is an important aspect that deserves attention. We have now incorporated this discussion into the manuscript, supported by relevant literature.
Your comment on the health-protective roles of polyphenols found in extra virgin olive oil and red wine—such as hydroxytyrosol and resveratrol—was also extremely helpful. These aspects have now been integrated into the appropriate sections of the manuscript (modifications are highlighted in blue). Additionally, we have included a dedicated subsection elaborating on the relationship between neurotrophic factors (NGF, BDNF) and the Mediterranean dietary pattern, with particular emphasis on the anti-inflammatory and neuroprotective effects of consuming vegetables in combination with EVOO.
Once again, thank you for your constructive comments and thoughtful suggestions—they have greatly contributed to the further enhancement of our manuscript.
With kind regards and sincere appreciation,
The Authors
Reviewer 3 Report
Comments and Suggestions for Authors
Lehoczki and co. conducted a narrative review exploring the adverse effects of the Western diet on cognitive and cerebrovascular health, with a particular focus on how these dietary patterns relate to traditional Hungarian eating habits and their implications for the design of the Semmelweis Study, a prospective cohort study aimed at promoting healthy aging in Hungary.
The abstract offers a broad overview but falls short of academic standards. It lacks a clear research objective and reads more like a general introduction than a focused summary. The structure is unclear, with background, findings, and study details mixed together, hindering readability. Crucial elements, such as a brief methodology description, are missing. Many statements are vague and lack concrete insights. The abstract should specify key findings (e.g., mechanisms linking the Western diet to cognitive decline), explain the literature selection process, and use clearer, more formal scientific language. Phrases like “highlighting the preventive potential” and “in Hungary and beyond” are awkward or informal for a scientific abstract. Overall, the abstract needs revision for clarity, precision, structure, and academic tone to effectively convey the review’s scope and significance.
The review effectively summarizes established associations between the Western diet and cognitive decline but offers limited new hypotheses, original mechanisms, or innovative perspectives. Although the manuscript claims to present a “Hungarian perspective,” much of the data and discussion remain generic and broadly applicable to global trends. The specific contextual value related to Hungary, such as cultural practices or regional interventions, is insufficiently explored or uniquely framed.
The Semmelweis Study is introduced as a planned cohort; however, critical methodological details are lacking. Recruitment criteria, sample size calculations, data collection instruments, and defined primary or secondary outcomes are not specified. Additionally, there is no information on statistical analysis plans, modeling approaches, power calculations, or descriptive analysis frameworks, significantly undermining the scientific rigor of this study component.
It is unclear how the study intends to isolate dietary effects from potential confounders like physical activity, socioeconomic status, or comorbidities. While epidemiological and cohort studies are referenced, the manuscript does not present original data, numerical analyses, or statistical validation of its claims. The review also falls short in critically addressing variability, confidence intervals, effect sizes, or the distinction between causality and correlation in the cited literature. Furthermore, the absence of a meta-analysis or systematic review methodology limits the strength and reliability of the conclusions drawn from existing studies.
Several sections—particularly those addressing mechanisms—are overly lengthy and descriptive, lacking critical evaluation of conflicting evidence or alternative explanations. The cited evidence is presented in a fragmented manner without cohesive synthesis or comparative summaries of epidemiological, clinical, and preclinical findings.
The absence of figures or visual aids is a notable weakness in a review with such an extensive narrative. Incorporating visual summaries, such as diagrams, pathways, or concept maps, would significantly enhance reader comprehension and scientific communication. Infographics illustrating mechanisms, comparative dietary patterns, or distinctions between the Hungarian and Western diets would add substantial value.
The discussion largely restates information already presented in the body of the manuscript, with minimal critical evaluation or integration of findings.
The authors do not attempt to assess the strength of evidence across different study types (e.g., epidemiological, preclinical, interventional). It also overlooks important limitations of the reviewed literature, such as confounding factors in observational studies, small sample sizes, and issues with generalizability.
Furthermore, the authors do not acknowledge potential limitations in their own review process, including possible selection bias or an incomplete search strategy. Without addressing these limitations, the conclusions lack necessary credibility and balance.
The connection between the review findings and the design or implementation of the Semmelweis Study remains unclear. There is no explanation of how the findings will guide the selection of variables, outcomes to be measured, or intervention strategies.
Additionally, the discussion fails to engage with conflicting or inconclusive evidence present in the literature on diet and cognition. For instance, some meta-analyses report modest or inconsistent effects of diet on cognitive outcomes, yet these important nuances are omitted from the discussion.
The reference list requires updating and expansion to more comprehensively reflect current work in the field. Journal names should follow the standard Index Medicus abbreviations. I recommend including the DOI for each referenced article to ensure easy access and accurate citation.
Comments on the Quality of English LanguageThe manuscript is mostly readable, but numerous minor issues affect its clarity and professional polish. Key problems include:
- Line 24 - “highlighting the preventive potential…”, should be: “emphasizing its potential for prevention...”
- Line 62 - “in shaping the trajectory of age-related cognitive decline”, should be: “in contributing to cognitive decline...”
- Line 75 - “processed foods, high intake of saturated fats…”, should be: “processed foods, saturated fats, and refined sugars, while being low in fruits…”
- Line 286 - “strategies may provide a window of opportunity…”, should be: “may offer a critical intervention period...”
- Line 414 - “Fish consumption increases the levels…”, should be: “increases plasma omega-3 fatty acids levels…”
- Line 609 - “hyper-palatability” is not a widely recognized academic term, should be defined or replaced
English including grammar, style and syntax, should be improved through the professional help from English Editing Company for Scientific Writings.
Author Response
Dear Reviewer,
We would like to sincerely thank you for your detailed comments, which have greatly helped us improve the manuscript.
Regarding the abstract, we acknowledge that it previously lacked a clear research objective, a structured format, and a methodological description. Accordingly, we have revised the abstract to highlight the mechanistic links between the Western diet and cognitive decline, provided a more detailed explanation of the literature selection criteria, and aimed to formalize the scientific language. Separating the results and the study’s relevance has also improved readability.
The manuscript does not aim to propose new hypotheses or present innovative mechanisms, but rather to synthesize existing scientific findings with special attention to dietary and lifestyle characteristics specific to the Hungarian population. We have specified aspects of the Hungarian context in more detail (e.g., obesity, chronic diseases, etc.), and emphasized the role of the Semmelweis Study as a domestic cohort investigation, along with the Semmelweis–EUniWell Health Promotion Model Program.
Regarding methodological concerns, we would like to emphasize that this manuscript is a theoretical and literature-based review intended to establish the professional framework for the Semmelweis Study. The publication of empirical data and statistical analyses is planned for the future.
To address the lack of figures and visual aids, we have added a new figure and a table to visually support the content overview and comprehension.
We have enhanced the discussion section by including a more critical integration and presentation of conflicting evidence, thus more clearly reflecting the boundaries of scientific consensus and the need for further research.
The reference list is now provided in two formats (APA and Vancouver) according to scientific conventions, and DOI identifiers have been added for more accurate referencing. We do not consider additional literature inclusion necessary at this stage (the manuscript already includes nearly 500 references), as the current literature base is already comprehensive and relevant.
We have thoroughly reviewed and improved the English language throughout the manuscript, fully addressing the grammatical and stylistic issues you identified, thereby raising the professional standard.
Once again, we thank you for your detailed and constructive suggestions, which have significantly contributed to enhancing the quality of our manuscript.
Respectfully,
The Authors
Round 2
Reviewer 2 Report
Comments and Suggestions for Authors
The paper is improved. However, some issues should be corrected.
-As for red wine drinking, the authors should write moderate red wine consumption, also in the abstract.
-The text should be better clarified so that the authors wrote a “narrative” review.
-The abbreviations should be written in full the first time they appear in the text, also in the abstract.
-The Latin name of the species should be written in Italics every time it appears in the text.
-The abstract could be shortened.
-Please update the abbreviation list.
-Do the arrows in Figure 1 indicate “increase/decrease”? If yes, please update the caption.
Comments on the Quality of English LanguageThe English could be improved to convey the research more clearly.
Author Response
Thank you very much for your thoughtful comments and constructive feedback!
We have carefully considered each of the points raised and have made all the suggested revisions to the manuscript:
- We have updated the term to "moderate red wine consumption," including in the abstract,
- We have clarified that the manuscript is a narrative review,
- All abbreviations are now written in full at their first appearance, both in the main text and the abstract,
- Latin species names are now consistently italicized throughout the manuscript,
- The abstract has been shortened for conciseness,
- The list of abbreviations has been updated accordingly,
- We have revised the caption of Figure 1 to clarify that the arrows indicate "increase" or "decrease."
Thank you once again for your careful review and valuable suggestions!
With kind regards,
The Authors
Reviewer 3 Report
Comments and Suggestions for Authors
The authors have significantly addressed the majority of the comments and suggestions provided, and the manuscript has been revised to reflect these changes. Based on the revisions, I believe the manuscript is now suitable for publication in this journal.
Comments on the Quality of English LanguageI recommend that the manuscript undergo a final review by a native English speaker to ensure the language is polished and free from any grammatical or stylistic issues that may affect clarity and readability.
Author Response
Dear Reviewer,
Thank you very much for your positive feedback and for your valuable contribution to the development of our manuscript! We are pleased to hear that, following the revisions, the manuscript is now considered suitable for publication. Once again, we sincerely appreciate your support and thorough review.
With kind regards,
The Authors